# New Acylcarnitine Ratio as a Reliable Indicator of Long-Chain 3-Hydroxyacyl-CoA Dehydrogenase Deficiency

**DOI:** 10.3390/ijns9030048

**Published:** 2023-08-25

**Authors:** Galina V. Baydakova, Polina G. Tsygankova, Natalia L. Pechatnikova, Olga A. Bazhanova, Yana D. Nazarenko, Ekaterina Y. Zakharova

**Affiliations:** 1Research Centre for Medical Genetics, Moskvorechye Str., 1, 115522 Moscow, Russia; 2Morozov Children’s City Clinical Hospital, 4th Dobryninsky Lane, 1/9, 119049 Moscow, Russia

**Keywords:** biochemical marker, inherited metabolic diseases, acylcarnitines, long-chain 3-hydroxyacyl-CoA dehydrogenase deficiency, *HADHA*

## Abstract

Long-chain 3-hydroxyacyl-CoA dehydrogenase (LCHAD) and mitochondrial trifunctional protein (MTP) deficiencies are rare fatal disorders of fatty acid β-oxidation with no apparent genotype–phenotype correlation. The measurement of acylcarnitines by MS/MS is a current diagnostic workup in these disorders. Nevertheless, false-positive and false-negative results have been reported, highlighting a necessity for more sensitive and specific biomarkers. This study included 54 patients with LCHAD/MTP deficiency that has been confirmed by biochemical and molecular methods. The analysis of acylcarnitines in dried blood spots was performed using ESI-MS/MS. The established “HADHA ratio” = (C16OH + C18OH + C18:1OH)/C0 was significantly elevated in all 54 affected individuals in comparison to the control group. Apart from 54 LCHAD deficiency patients, the “HADHA ratio” was calculated in 19 patients with very-long-chain acyl-CoA dehydrogenase (VLCAD) deficiency. As VLCAD-deficient patients did not show increased “HADHA ratio”, the results emphasized the high specificity of this new ratio. Therefore, the “HADHA ratio” was shown to be instrumental in improving the overall performance of MS/MS-based analysis of acylcarnitine levels in the diagnostics of LCHAD/MTP deficiencies. The ratio was demonstrated to increase the sensitivity and specificity of this method and reduce the chances of false-negative results.

## 1. Introduction

Long-chain 3-hydroxyacyl-CoA dehydrogenase (LCHAD) deficiency (OMIM# 609016) is a rare potentially fatal autosomal recessive disorder of fatty acid β-oxidation (FAO). LCHAD, long-chain enoyl-CoA hydratase (LCEH) and long-chain 3-ketoacyl-CoA thiolase (LCKAT) together constitute mitochondrial trifunctional protein (MTP). MTP is bound to the inner mitochondrial membrane and plays an essential role in the catalysis of the last three steps of long-chain FAO. LCEH and LCHAD are located in the N- and C-termini of the α-subunit of MTP, respectively, encoded by the *HADHA* gene. LCKAT is located in the N-terminus of the β-subunit of MTP encoded by the *HADHB* gene [1]. Both genes are localized on chromosome 2 (2p23). Reported mutations in *HADHA* or *HADHB* mostly affect the catalytic sites of MTP and commonly alter the activity of only one enzyme. Mutations affecting amino acid residues localized at the interface of the dimerization domains between α- and β-subunits of the MTP complex result in the reduced activities of all three enzymes and complete MTP deficiency (OMIM# 609015) [2].

The most frequent type of MTP defect is isolated LCHAD deficiency. It affects <1.5 per 100,000 births in prevailing European populations [3]. The disorder was first described in 1989 by Wanders and colleagues [4]. According to the literature, the pathogenic missense variant NM_000182.5: c.1528G > C (p.Glu510Gln) in exon 15 of the *HADHA* gene is considered the most common genetic variant, which is observed in 90% of European patients with LCHAD-deficient alleles [5]. To date, no strong association between the severity of the disease/mutation type and the clinical phenotype has been established [6].

Being a disorder of FAO, LCHAD/MTP deficiency is characterized by severe bioenergetic imbalance. Normally, FAO produces acetyl-coenzyme A (acetyl-CoA), reduced nicotinamide adenine dinucleotide (NADH) and reduced flavin adenine dinucleotide (FADH2). NADH and FADH2 molecules are transferred to the electron transport chain for adenosine triphosphate (ATP) generation, whereas acetyl-CoA is further converted to ketone bodies or is oxidized via the tricarboxylic acid cycle. Disrupted FAO in LCHAD-deficient individuals leads to a decrease in the production of ketone bodies and ATP, and therefore inadequate energy supply [7]. Furthermore, LCHAD/MTP deficiency leads to the accumulation of toxic FAO intermediates, thereby inducing oxidative stress, lipotoxicity and altering cell homeostasis.

Clinical severity and the age of LCHAD/MTP deficiency onset are highly variable. Clinical manifestations in the affected individuals are heterogeneous. Symptoms are triggered by long fasting, exercise, fever, and other causes of metabolic stress, such as surgery or trauma. Three main forms of LCHAD/MTP deficiencies have been reported: (1) severe, early-onset form, which is associated with cardiomyopathy, hypoglycemia and sudden infant death; (2) intermediate, infant-onset form, which is characterized by recurrent hypoketotic hypoglycemia and lethargy during illness or fasting; and (3) a milder, late-onset form that is triggered by exercise, fasting or infections and is associated with progressive peripheral neuropathy and recurrent rhabdomyolysis [1,8,9,10].

Seizures, speech and developmental delay and retinopathy are other features of LCHAD/MTP deficiencies [11]. Furthermore, the similarity in some symptoms (e.g., elevated lactic acid, cardiomyopathy, polyneuropathy, retinopathy) between LCHAD/MTP deficiencies and mitochondrial respiratory chain disorders may be explained by the fact that MTP participates in cardiolipin remodeling and physically interacts with mitochondrial respiratory chain complex 1 [12,13,14].

LCHAD/MTP-deficient individuals frequently have liver diseases (hepatic dysfunction, hepatomegaly, steatosis, etc.), especially during episodes of metabolic decompensation. Moreover, the release of toxic 3-hydroxy intermediate metabolites from the LCHAD/MTP-deficient placenta and fetus into the maternal circulation is likely to be a culprit in inducing Acute Fatty Liver of Pregnancy (AFLP) in the pregnant mother [15].

LCHAD/MTP deficiency therapy includes fasting avoidance as well as a diet restricted in long-chain fatty acids and supplemented with both medium-chain triglycerides (MCT) and essential fatty acids.

The earlier recognition and treatment of LCHAD/MTP deficiency are crucial to diminish its high morbidity and mortality due to metabolic decompensations, prompting the inclusion of LCHAD/MTP deficiency in newborn screening (NBS) programs. According to the study of den Boer and coauthors [16], 38% of cases ended up with death within the first three months after the disease onset.

The current diagnostic workup for LCHAD/MTP is mainly based on the measurement of acylcarnitines in dried blood spots (DBS) using tandem mass spectrometry (MS/MS) [17,18,19]. Eight different analytes or ratios are used as markers in screening for LCHAD/MTP deficiencies: 3-hydroxytetradecanoylcarnitine (C14OH), tetradecenoylcarnitine (C14:1), 3-hydroxypalmitoylcarnitine (C16OH), 3-hydroxypalmitoleylcarnitine (C16:1OH), 3-hydroxystearoylcarnitine (C18OH), 3-hydroxyoleoylcarnitine (C18:1OH), oleoylcarnitine (C18:1) and 3-hydroxypalmitoylcarnitine/palmitoylcarnitine (C16OH/C16) [8]. Nevertheless, false-positive results and thus overdiagnosis of LCHAD/MTP deficiency after acylcarnitine measurement have been reported [8,18,19].

Importantly, patients with LCHAD/MTP deficiencies commonly experience low levels of free carnitine (C0). Low levels of free carnitine, in turn, lead to a decrease in the levels of all acylcarnitines [20]. This decrease leads to false-negative results, reducing the sensitivity of these biomarkers. Indeed, several reports on a missed diagnosis of LCHAD/MTP deficiencies after neonatal screening by acylcarnitine measurement have been published [8,21,22].

Thus, we hypothesize that it is crucial to take into account free carnitine (C0) concentration when referring to concentrations of C18OH, C18:1OH and C16OH. The ratio of the sum of C18OH, C18:1OH and C16OH to C0 ((C16OH + C18OH + C18:1OH)/C0) named “HADHA ratio” was suggested to improve the sensitivity and specificity of the diagnostic method.

## 2. Materials and Methods

### 2.1. Study Participants

The study included 54 patients (18 females and 36 males) with LCHAD deficiency confirmed by biochemical and genetic methods since 2004. The mean age at diagnosis was 16.3 months (SD 48.4; age range = 4 days–26 years). In addition, 19 patients (12 females and 7 males) with very-long-chain acyl-CoA dehydrogenase (VLCAD) deficiency, confirmed by biochemical and genetic methods since 2006, were included in the study to evaluate the specificity of the “HADHA ratio”. The mean age at diagnosis was 96.1 months (SD 195; age range = 2 days–57.5 years). All samples had been deposited at the Moscow Branch of the Biobank “All-Russian Collection of Biological Samples of Hereditary Diseases” (Research Centre for Medical Genetics, Moscow, Russia).

The control group consisted of 500 individuals (313 males and 187 females) who had been referred to Research Centre for Medical Genetics (Moscow, Russia) for genetic diagnosis since 2004 and in whom no inborn errors of metabolism had been diagnosed. The mean age of the control group was 31.6 months (SD 26.2; age range = 8 days–7.5 years). The range of age of the control group was consistent with the range of age at diagnosis in the patients.

All patients provided informed consent for participation in the study and deposition of their biological samples in the Biobank. The study followed the guidelines of the ethics committee of the Research Centre for Medical Genetics (Moscow, Russia).

### 2.2. Sample Preparation

For most cases, biochemical and DNA analysis was performed using only dried blood spot (DBS) samples (due to their availability), although several subjects had both EDTA whole blood and DBS samples. The DBS were collected using standard procedures by puncturing the fingertip with a lancet. The collected whole blood was dropped onto a 903 sample collection card (PerkinElmer^®^; Waltham, MA, USA) and then air-dried for approximately 2 h. The samples collected as EDTA whole blood were spotted onto the filter paper using 50 μL of the whole blood. One disk (3.2 mm) was cut from each DBS sample with a DBS puncher (PerkinElmer^®^; Waltham, MA, USA).

### 2.3. Biomarker Analysis

The ESI-MS/MS analysis of acylcarnitines was performed using NeoGram AAAC MassSpectrometry kit (Perkin Elmer^®^, Waltham, MA, USA) and AB SCIEX 3200 Qtrap (Sciex, Redwood City, CA, USA).

### 2.4. Genotyping

Genomic DNA was extracted from the whole blood samples with EDTA and DBS using GeneJET Genomic DNA Purification Kit (Thermo Fisher Scientific, Waltham, MA, USA) according to the manufacturer’s protocol. The exons 1–20 of the *HADHA* gene and the adjacent intronic sequences as well as the exons 1–20 of the *VLCAD* gene and the adjacent intronic sequences were analyzed using Sanger sequencing (the primer sequences are available upon request) on ABI PRISM 3500×L Genetic Analyzer (Applied Biosystems, Foster City, CA, USA).

### 2.5. Statistical Analysis

Statistical analysis was performed using GraphPad Prism 8.0.1 software (GraphPad Software Inc., San Diego, CA, USA). The nonparametric Mann–Whitney test was used to determine the statistical differences between the groups. If required, the statistical significance was assessed using multiple comparisons. A *p*-value < 0.05 was considered statistically significant. Receiver operating characteristic (ROC) curves were allowed to define the cut-off value, and the area under the curve (AUC) was used to determine the diagnostic accuracy of the test. Reference intervals (0.1th and 99.9th percentile) were calculated.

## 3. Results and Discussion

Acylcarnitine profile measurement using MS/MS remains a gold standard in the diagnosis of FAO disorders [23,24,25,26]. According to our experience and the literature data, several variations of acylcarnitines can be detected in LCHAD/MTP-deficient patients:

Group 1. Classic acylcarnitine profile, which is usually observed during metabolic decompensation, with an increase in long-chain 3-hydroxyacylcarnitines (C14OH, C16OH, C16:1OH, C18OH and C18:1OH) and with normal C0 level. It is usually observed in patients with chronic LCHAD/MTP deficiency as well as in the preclinical stage of the disease.

Group 2. Classic acylcarnitine profile with a change in the level of one or two long-chain 3-hydroxyacylcarnitines (usually C16OH or/and C18OH) and a decreased level of C0. It is usually observed in symptomatic patients, in newborn screening program.

Group 3. Severely altered acylcarnitine profile with an increase in the concentration of long-chain 3-hydroxyacylcarnitines and most of the long-chain acylcarnitines (C14OH, C14, C14:1, C16OH, C16:1OH, C18OH, C18:1 and C18:1OH) as well as a decreased/normal C0 level, which is observed during an acute metabolic crisis.

Group 4. Acylcarnitine profile with a decreased C0 level and with blood acylcarnitines at the upper limit of normal. It is usually observed in patients with chronic LCHAD/MTP deficiency, and in some patients with intermediate form.

For Group 3, differential diagnosis with other FAO disorders may be challenging, and Group 4 may be misdiagnosed.

Since the increase in C18OH, C18:1OH and C16OH concentrations serves as the main biochemical marker in the diagnostics of LCHAD/MTP deficiency, and given the fact that a great number of patients with these deficiencies have low levels of free carnitine (C0), it was suggested that the ratio of the sum of C18OH, C18:1OH and C16OH to free carnitine ((C16OH + C18OH + C18:1OH)/C0) may improve the sensitivity of the method.

In this research, 54 patients with genetically verified LCHAD deficiency (pathogenic variants in the *HADHA* gene) and 500 samples from the control group were analyzed. The “HADHA ratio” = (C16OH + C18OH + C18:1OH)/C0 was calculated to evaluate its specificity and sensitivity in the LCHAD deficiency diagnostics (Appendix A). It should be noticed that Patient 17 demonstrated significantly higher acylcarnitines and free carnitine values in comparison to other patients due to carnitine supplementation received in the intensive care unit.

Importantly, in this research, 39% of the patients with LCHAD deficiency had low levels of free carnitine. Classic acylcarnitine profile with normal C0 (Group 1) was observed in 14 patients (26%). Classic acylcarnitine profile with a decreased level of C0 (Group 2) was observed in 13 patients (24%). A severely altered acylcarnitine profile with an increase in the concentration of long-chain 3-hydroxyacylcarnitines and most of the long-chain acylcarnitines (Group 3) was observed in 24 patients (44%). Finally, acylcarnitine profile with a decreased C0 level and normal long-chain 3-hydroxyacylcarnitines (Group 4) was observed in three patients (6%).

As Groups 1 and 2 patients have elevated acylcarnitines characteristic of LCHAD deficiency, the diagnosis is straightforward.

The diagnosis of LCHAD deficiency in Group 3 may pose a challenge, since in addition to the acylcarnitines characteristic of LCHAD deficiency, the affected individuals have elevated levels of acylcarnitines characteristic of very-long-chain acyl-CoA dehydrogenase (VLCAD) deficiency. Thus, in this research, the “HADHA” ratio of 0.005 ± 0.004 (SD) was calculated in 19 patients with VLCAD deficiency (vs. 0.19 ± 0.14 (SD) in LCHAD/MTP deficiency patients). According to the nonparametric Mann–Whitney test, there was a significant difference between these groups (*p* < 0.0001), differentiating these two conditions. These results emphasize the high specificity of this new “HADHA” ratio.

The greatest difficulty in diagnosis of LCHAD deficiency is observed in Group 4 patients since in these individuals acylcarnitines that are characteristic of LCHAD deficiency are within the reference intervals due to low levels of C0 (and therefore decreased levels of all acylcarnitines). Importantly, the “HADHA ratio” allows us to diagnose LCHAD deficiency in Group 4 patients, since even though characteristic acylcarnitines are normal, “HADHA ratio” is elevated (Table 1).

The graphs of acylcarnitine ratios for the LCHAD deficiency were presented according to the research of McHugh and coauthors [26]. The following results (patients vs. control group) were obtained: the ratio C16OH/C16 of 0.447 ± 0.214 vs. 0.022 ± 0.019 (Figure 1A), and the “HADHA ratio” of 0.19 ± 0.14 vs. 0.0023 ± 0.0016 (Figure 1B). The largest AUC was calculated (Table 2).

We used the cut-off value of 0.1 C16OH/C16 ratio from the research of McHugh and coauthors [26] in our study cohort, as shown in Figure 1. The cut-off value = 0.1 demonstrated Se = 98.2% and Sp = 98.8% (vs. Se = 100% and Sp = 100% of the “HADHA ratio”) (*p* < 0.001).

Hence, both ratios were shown to be efficient in LCHAD deficiency diagnostics, differentiating the affected individuals from a control group. Nevertheless, the new “HADHA ratio” demonstrated higher specificity in comparison to C16OH/C16. Furthermore, our study cohort included one patient (Patient 11) with the concentration of C16OH = 0 (Appendix A) and therefore C16OH/C16 = 0.

In this research, the cut-off value for the established “HADHA ratio” determined by the ROC curve analysis was ≈0.027 (Table 3). The ratio has shown diagnostic efficiency at low concentrations of C0 (the minimum value in the study cohort was 0.03), thus demonstrating high analytical performance for LCHAD/MTP deficiencies. For this reason, the established novel “HADHA ratio” will allow differentiating patients with LCHAD/MTP deficiency from a control group and increase the specificity of the MS/MS-based measurement of acylcarnitine levels.

Although not the main objective of this paper, the first symptoms, the age at LCHAD deficiency onset, the age at death, main clinical symptoms, phenotype (according to the age of onset and disease severity) and family history of the affected individuals were also recorded. The mean age at diagnosis was 16.3 months (SD 48.4; age range = 4 days–26 years). The earliest diagnosis was conducted in a 4-day-old patient, and the latest diagnosis was conducted in a 26-year-old patient. The main clinical symptoms of the study cohort included hypoglycemia, cardiomyopathy, hepatic impairment (i.e., hepatomegaly and hepatic cytolysis), hypotonia and cramps. A total of 12 patients (22%) had severe LCHAD deficiency phenotype and developed the first symptoms during the first days of life. Furthermore, 37 patients (68.5%) had intermediate phenotypes precipitated by infection and developed the first symptoms during the first months of life. Three patients had mild LCHAD deficiency phenotype. Patients 10 and 48 were asymptomatic (the diagnosis was conducted using neonatal screening) and received therapy in the preclinical stage of the disease. All available clinical data are summarized in Appendix A.

## 4. Conclusions

Analysis of acylcarnitine profile via MS/MS has been a gold standard approach for FAO disorders, although its sensitivity and specificity remain insufficient. The use of ratios of acylcarnitines improves the sensitivity of the method. For example, in VLCAD deficiency, the ratio of tetradecenoylcarnitine (C14:1) over acetylcarnitine (C2) results in less false-negative results when compared to the use of C14:1 as the only marker. Meanwhile, C16OH/C16 is the most common ratio used for diagnostics of LCHAD/MTP deficiencies, avoiding challenges associated with variability in the cut-offs between studies as well as differences in the extraction and calibration methods.

Measurement of the established “HADHA ratio” = (C16OH + C18OH + C18:1OH)/C0 demonstrated 100% specificity and 100% sensitivity in screening for LCHAD/MTP deficiencies. As the results demonstrated, the “HADHA ratio” may remarkably improve overall analytical performance and the detection rate of suspected LCHAD/MTP deficiencies, reducing the chances of false-negative results and ruling out other conditions associated with low C0 and acylcarnitines levels. Moreover, as VLCAD-deficient patients did not show an increased “HADHA ratio”, the proposed ratio may differentiate these two disorders.

The main limitation of this research was the limited number of newborns in the study cohort. Being an extremely rare disorder with a broad phenotypic spectrum, the age of LCHAD/MTP deficiency onset is highly variable, which explains the wide age range of this study cohort. Nevertheless, follow-up research on a larger cohort of newborns is crucial to emphasize the sensitivity of the new “HADHA” ratio in screening for LCHAD/MTP deficiency.

## Figures and Tables

**Figure 1 IJNS-09-00048-f001:**
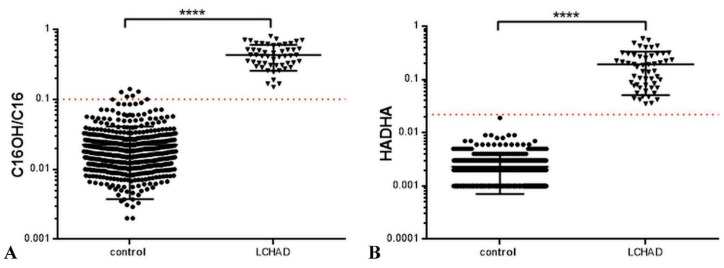
(**A**) The ratio C16OH/C16 of the patients diagnosed with LCHAD deficiency and the control samples, **** *p* < 0.0001; (**B**) The “HADHA ratio” = (C16OH + C18OH + C18:1OH)/C0 of the patients diagnosed with LCHAD deficiencies and the control samples, **** *p* < 0.0001. Circles indicate control group subjects; triangles—LCHAD deficiency patients; red-pointed line—the cut-off value.

**Table 1 IJNS-09-00048-t001:** Acylcarnitine concentrations (μmol/L, mean ± SD) and their ratios in Groups 1–4.

	C0	C14:1	C16:1OH	C16OH	C18:1OH	C18OH	“HADHA Ratio”	C16OH/C16
1	4.4 ± 1.8	0.21 ± 0.11	0.14 ± 0.09	0.32 ± 0.20	0.49 ± 0.25	0.37 ± 0.15	0.29 ± 0.14	0.43 ± 0.19
2	17.5 ± 8.5	0.22 ± 0.09	0.12 ± 0.11	0.45 ± 0.3	0.64 ± 0.77	0.5 ± 0.31	0.1 ± 0.1	0.39 ± 0.15
3	24.0 ± 18.4	0.68 ± 0.3	0.31 ± 0.14	1.03 ± 0.51	1.05 ± 0.57	0.99 ± 0.62	0.21 ± 0.13	0.48 ± 0.16
4	5.3 ± 2.2	0.08 ± 0.05	0.03 ± 0.03	0.08 ± 0.01	0.09 ± 0.03	0.1 ± 0.05	0.06 ± 0.01	0.21 ± 0.07
RI	8–90	<0.32	<0.22	<0.18	<0.16	<0.15	<0.027	<0.1 *

Symbols and abbreviations are as follows: “HADHA ratio” = (C16OH + C18OH + C18:1OH)/C0; RI = reference intervals; * = the value is taken from McHugh and coauthors [26].

**Table 2 IJNS-09-00048-t002:** The area under the curve (AUC) of the estimated ratios or acylcarnitines.

AC or Ratio	AUC
C16OH	0.981
C16:1OH	0.851
C18OH	0.999
C18:1OH	0.999
C16OH/C16	0.982
“HADHA Ratio”	1

AC = acylcarnitine; AUC = area under the curve; “HADHA ratio” = (C16OH + C18OH + C18:1OH)/C0.

**Table 3 IJNS-09-00048-t003:** The determination of the cut-off for the new “HADHA ratio”.

Cut-Off	Sensitivity (%)	95% CI	Specificity (%)	95% CI
>0.0085	100	93.4% to 100%	99.0	97.68% to 99.67%
>0.014	100	93.4% to 100%	99.8	98.89% to 99.99%
>0.027	100	93.4% to 100%	100	99.26% to 100%
>0.036	98.2	90.11% to 99.95%	100	99.26% to 100%
>0.039	96.3	87.25% to 99.55%	100	99.26% to 100%

## Data Availability

Not applicable.

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
