# Peer review of "New Acylcarnitine Ratio as a Reliable Indicator of Long-Chain 3-Hydroxyacyl-CoA Dehydrogenase Deficiency"

_2409-515X, 2023, doi:10.3390/ijns9030048_

Round 1

Reviewer 1 Report (Previous Reviewer 4)

The paper looks more complete and informative than the previous version. 

Authors made a satisfactory work. It is ok for me.

Reviewer 2 Report (Previous Reviewer 5)

Thank you for asking me to review this manuscript, which is a revised version of a manuscript previously submitted and reviewed (IJNS 2340566). The revised manuscript highlights the additions/amendments but I do not see reference to the previous Reviewers' comments.

The revised manuscript appropriately describes the development of the use of the acylcarnitine ratio calculation "HADHA ratio" to improve the diagnostic performance in detecting patients with LCHAD/MTP deficiency. The revision has addressed the previous comments I had made including description of the two genes involved.

I am satisfied with the amendments made and would recommend accepting this manuscript.

This manuscript is a resubmission of an earlier submission. The following is a list of the peer review reports and author responses from that submission.

Round 1

Reviewer 1 Report

The manuscript under appreciation studies New Acylcarnitine Ratio as a Reliable Indicator of Long-Chain 3-Hydroxyacyl-CoA Dehydrogenase Deficiency.

In my opinion, this manuscript is not suitable for publication. In the introduction section, authors didn't explain well why they propose this new ratio. There is incorrect information. Also, literature review and references needed to be improved. Good papers are missing (instead citing a PhD thesis). In addition, there is a lot of information missing in the methods part. The results and discussion part is also very confusing.

Reviewer 2 Report

The authors established a method to analyze the acylcarnitine profile via MS/MS, which is useful for detection fatty acid oxidation disorders caused by LCHAD/MTP deficiencies. The authors perform a cohort of LCHAD deficiency group and analyzed  the ratio C16OH/C16 of those patients. This study contributes a new method to diagnosis for FAO disorders. However, the data is thin, and need to show a more comprehensive information. 

1. For 54 patients, please show what kind of LCHAD deficiency for each patient, what is the pathological phenotype if available, such as cardiomyopathy, hypoglycemia, or liver failure. 

2. It is better to analyze LCHAD or related protein level in plasma if available, or other proteomics, lipidomics, metabolomics. This will give a comprehensive profile for the patients. 

Reviewer 3 Report

Comments and suggestions in the file.

Reviewer 4 Report

Authors show the results of a study conducted to evaluate the diagnostic value of a new acylcarnitine ratio that could lead to a more sensitive and specific diagnosis of LCHAD/MTP deficiency. The topic is of importance in the practice of extended newborn screening, because it aims to improve the diagnosis of one of the most severe beta oxidation disorders, still burdened by negative outcome. However, there are some elements that must be improved, especially concerning the information provided on the study participants, which are completely missing and represent an important element potentially implicated in the interpretation of the results.

In detail:

Abstract

No information are given on control group, please add basic info in the abstract, in order to characterize the matching group.

Introduction

Line 50-51 Authors state: “Importantly, the affected neonates with asymptomatic or mitigated phenotypes may be 50 fatally misdiagnosed by neonatal screening”. In neonatal screening newborns are asymptomatic by definition, please rewrite this sentence.

Line 62-63”…as well as a partial verification bias 62 in acylcarnitines measurement using MS/MS”: no clear sentence, please clarify

Materials and methods

Study participants: why age is not reported with common measures of central tendency?  it would have resulted easier to compare this variable between patients and control group. Range age is wide perticularly for controls. What is about the timing of diagnostic test? Was it performed at neonatal age for everyone or later? Author state that the oldest controls are aged    years. Was the test available 28 years ago. Please expand on that.

Moreover, apart from age, no other information are reported about study participants: gender, gestational age, admission in ICU, type of feeding, all important covariates to be considered for the diagnosis of LCHAD as they can alter the acylcarnitine profile and be the cause of false positive and negative, as the same authors claim in the text.

Also, although no clear genotype phenotyp correlation is so far available, it would be useful to have some information on genetic tests results for patients even as supplementary material.

Statistical analysis Authors state that non parametric tests have been used, nevertheless the high numerosity of the control group lead to hypothisize a normal distribution of variables. Why parametric tests have not been used?

Results and Discussion

Potential advantages of the research are emphasized, please add limitations of the study research and future potential methodology improvement.

Reviewer 5 Report

Thank you for asking me to review this manuscript, which describes utilising the ratio of specified long-chain acylcarnitines to free carnitine as a way to improve the diagnostic power of the acylcarnitine profile in detecting LCHAD/MTP deficiency.

Introduction: It would be helpful to include description of the role of the HADHB gene as well as HADHA in forming the LCHAD/MTP complex. In description of the pathogenesis of the associated disorder, it would also be important to mention the impact of the energy defect generated by abnormal beta-oxidation, and not just the toxic effect of the long chain fatty acids/ acylcarnitines.

Page 2 line 51 - "fatally misdiagnosed " - what does this mean? Are these patients missed by screening, or misclassified as having severe disease?

The introduction needs to include more detail on the hypothesis that the "HADHA" ratio could be an improved method for diagnosis. (Paragraph on page 3 line 127-132 could be moved to the introduction).

Methods:

The study included 54 patients, aged 16.3+/-48.4 months (range 3days -26 years). What does 16.3 months represent- is this mean (or median) +/- standard deviation?

Need to specify who the control subjects were. Are these healthy volunteers? 

Samples: did all patients and controls have a DBS and EDTA whole blood sample obtained prospectively as part of this study, or were historic samples used for some?

Genotyping: was this done for patients and controls? I note that some of the genotype data was incomplete in Supplementary table S1 with some patients only having a single heterozygous variant identified, and explained because "unidentified mutation due to a patient’s refusal to proceed the research"

This needs to be explained-  did these patients not consent to be included? Or withdraw consent part way through?

The methods suggest only HADHA was sequenced. Were any patients found with HADHB variants?

Results/Discussion

These sections are merged. It would be preferable to have results outlined first followed by the discussion.

The result presented here shows that both C16OH/C16 and the "HADHA ratio" both achieve statistically significant differentiation of the control group from the patient group, with improved Sensitivity and Specificity for the HADHA ratio when comparing LCHAD/MTP to healthy controls.

The discussion mentions the other long chain fatty acid oxidation disorders that can also have abnormal acylcarnitine profiles. It would be helpful to have samples from patients affected with VLCADD, CPTII etc as part of the analysis to demonstrate whether the method described can actually differentiate LCHAD/MTP from these other similar disorders; this is important in confirming the Specificity of the test (as opposed to just testing it against a healthy control population).

The authors do acknowledge the potential differences in the neonatal period, and implications for whether this method could be employed in newborn screening. As the sample population range to many years of age, this cannot be directly extrapolated.